# New Records of Wood- and Bark-Inhabiting Nematodes from Woody Plants with a Description of *Bursaphelenchus zvyagintsevi* sp. n. (Aphelenchoididae: Parasitaphelenchinae) from Russia

**DOI:** 10.3390/plants12020382

**Published:** 2023-01-13

**Authors:** Alexander Y. Ryss, Sergei A. Subbotin

**Affiliations:** 1Zoological Institute, Russian Academy of Sciences, Universitetskaya Emb. 1, 199034 Saint Petersburg, Russia; alryss@gmail.com; 2Plant Pest Diagnostic Centre, California Department of Food and Agriculture, 3294 Meadowview Road, Sacramento, CA 95832, USA; 3Department of Entomology and Nematology, Hutchison Hall, University of California, Davis, CA 95616, USA; 4Centre of Parasitology, A.N. Severtsov Institute of Ecology and Evolution, Russian Academy of Sciences, Leninskii Prospect 33, 117071 Moscow, Russia

**Keywords:** *Aphelenchoides*, bark beetles, *Bursaphelenchus*, *Cryptaphelenchus*, *Deladenus*, *Diplogasteroides nix*, *Laimaphelenchus*, new species, 28S rRNA gene

## Abstract

Wood- and bark-inhabiting parasitic nematodes are of great economic importance. Nematodes can cause wilt diseases in conifers and deciduous trees. In 2014–2022, during nematology surveys conducted in different regions of Russia and Belarus, adults and dauer juveniles of nematodes were collected from wood, bark and beetle vectors. Using traditional morphological taxonomic characters integrated with molecular criteria, we identified in the studied samples the following nematode species: *Aphelenchoides heidelbergi*, *Bursaphelenchus eremus*, *B. fraudulentus*, *B. michalskii*, *B. mucronatus*, *B. willibaldi*, *Deladenus posteroporus*, *Diplogasteroides nix* and *Laimaphelenchus hyrcanus*, several unidentified species: *Aphelenchoides* sp.1 and sp.2, *Cryptaphelenchus* sp.1, sp.2 and sp.3, *Laimaphelenchus* sp.1, *Micoletzkya* sp.1, *Parasitaphelenchus* sp.1, *Parasitorhabditis* sp.1, three unidentified tylenchid nematodes and a new species, *Bursaphelenchus zvyagintsevi* sp.n. Morphological descriptions and molecular characterization are provided for *B. zvyagintsevi* sp. n. belonging to the *Abietinus* group and *B. michalskii* belonging to the *Eggersi* group. Findings of *Aphelenchoides heidelbergi*, *Bursaphelenchus eremus*, *B. michalskii*, *Deladenus posteroporus*, *Diplogasteroides nix* and *Laimaphelenchus hyrcanus* are new records for Russia. Phylogenetic positions of studied species were reconstructed using D2–D3 expansion segments of 28S rRNA gene sequence analysis. The data obtained in this study may help to detect the refugia of opportunistic plant pests and find possible native biocontrol nematode agents of insect vectors causing diseases.

## 1. Introduction

Wood- and bark-inhabiting parasitic nematodes cause wilt and dieback diseases in conifers and deciduous trees. Their destructive impact on forestry leads to significant economic and social consequences worldwide. Protective measures against nematodes are obligatory in all countries. Due to the prohibition or restriction of many nematicides, alternative control strategies are needed to control these diseases. Accurate nematode diagnostics, early detection of the neglected nematode pest refugia, identification of insect vectors of nematodes and suppressive bacterial and fungal agents are important for the development of effective control measures [1,2,3,4,5].

The aim of the present study was to analyze the data on the systematics and distribution of wood- and bark-inhabiting nematodes collected in Russia and Belarus from 2014 to 2022 using an integrative approach based on analyses of the molecular and morphological characters of parasitic nematodes extracted from wood and bark samples. The main goal of the study was to identify the wood- and bark-inhabiting parasitic nematodes collected in natural forest refugia in Russia, especially in the Siberian and Pacific regions, where numerous opportunistic pathogens associated with insect vectors, fungi and bacteria are found in weak and dying old trees in forest ecosystems. In this study, the tree wood and bark samples collected from ecosystems under anthropogenic transformations in urban areas, parks and botanical gardens were also included in the analysis. Under quickly changing climatic and biotic conditions and due to the anthropogenic disturbances of forest ecosystems, these nematodes may change their hosts and vectors and transform into the true pathogens [6,7,8,9].

In Russia, wood- and bark-inhabiting nematodes have been intensively studied and lists of associated species have been reported in several publications [10,11,12,13,14,15,16,17,18,19,20,21,22,23,24,25,26,27,28,29,30]. However, with the exception of *Bursaphelenchus*, the identification of species was based on morphology only and, therefore, now requires molecular confirmation.

The data presented in this study may aid in detections of refuges of opportunistic plant pests and possible native biocontrol nematode agents of insect vectors of diseases.

## 2. Results and Discussion

### 2.1. Species Identification and Ecological Grouping

Using traditional morphological taxonomic characters integrated with molecular criteria, we distinguished the following valid known nematode species that were found in the studied samples: *Aphelenchoides heidelbergi* (Zhao et al., 2007) Carta et al., 2016, *Bursaphelenchus eremus* Rühm (1956), *B. fraudulentus* (Rühm, 1956) Goodey 1960, *B. michalskii* Tomalak and Filipiak, 2018, *B. mucronatus* Mamiya and Enda, 1979, *B. willibaldi* Schönfeld et al., 2006, *Deladenus posteroporus* Yu et al., 2017, *Diplogasteroides nix* Kanzaki et al., 2016, *Laimaphelenchus hyrcanus* Miraeiz et al., 2015 and several unidentified species: *Aphelenchoides* sp.1 and sp.2, *Cryptaphelenchus* sp.1, sp.2 and sp.3, *Laimaphelenchus* sp.1, *Micoletzkya* sp.1, *Parasitaphelenchus* sp.1, *Parasitorhabditis* sp.1, *Sphaerulariid nematode* sp.1, *Tylenchid nematode* sp.1 and sp.2. and a new species, *Bursaphelenchus zvyagintsevi* sp. n. (Table 1).

After consideration of the biology of nematodes associated with bark beetles, Polyanina et al. [31] revealed several types of relations in the nematode–beetle–tree associations. Following this classification, several ecological groups of nematodes associated with bark beetles could be distinguished among the studied samples: obligatory ectophoronts, fungal and plant feeders (*Bursaphelenchus*, *Cryptaphelenchus*, *Aphelenchoides*); facultative phoronts and fungivores (*Laimaphelenchus*); facultative phoronts, bacterial and fungal spore feeders (*Diplogasteroides*); facultative phoronts and carnivores (*Micoletzkya*); gut bark beetles endoparasites (*Parasitorhabditis*); hemocoel parasites of larval insect stage (*Parasitaphelenchus*); hemocoel parasites of adult insect stage (*Deladenus* and sphaerulariid nematodes).

#### *Bursaphelenchus zvyagintsevi* sp. n.

*Adults* (Figure 1 and Figure 2, Table 2): http://zoobank.org/urn:lsid:zoobank.org:act:71BDE14D-B778-40FE-977D-D910F701DE6D (accessed on 13 August 2023). Body curved ventrally. Stylet base slightly expanded, but without distinct knobs. Cephalic annuli faintly distinct through light microscopy. Median bulb ellipsoid, large; valve median to sub-median of bulb. Excretory pore located at nerve ring or at posterior end of the median bulb. Lateral field with two incisures.

*Male*: Similar to female in structure of anterior end. Testis situated on right subventral side of mid-intestine, long, anteriorly reflexed, zone of spermatids distinct, consisting of one or two quartets of large, separated cells, and a zone of granulated immature sperm cells located posterior to spermatids. Sperm gradually decreasing in size to spherical mature sperm cells filling posterior 20–25% part of testis. Tail strongly hooked, J-shaped, terminating with very small bursa 6–8 μm at borders, and 3–5 μm along mid-line. Bursa posterior edge truncate to conical. There are seven male tail papillae: mid-ventral unpaired P1 just anterior to cloacal opening, paired P2 almost at same level as P1 laterally (0–13% of tail length), paired P3 shifted to 50–59% of tail length from cloacal opening, and paired, small, pore-like P5 close to ventral mid-line at level of lateral edges of bursa 81 (74–86%). P5 pair may be considered ‘gland papillae’ because of their pore-like form, whereas other papillae are nipple-like. P4 pair absent. Spicule strong, narrow, J-shaped, rostrum and condylus well developed and separated. Angle between lines along capitulum (condylus–rostrum) and extending spicule end = 13–23°, point of intersection ventral. Rostrum rounded to obtuse angular. Junction of rostrum and calomus rectangular. Condylus forms a round dorsal protrusion from the lamina curve, sometimes angular. Capitulum of spicule almost straight, flattened, sometimes with a shallow depression. Ventral velum present. Spicular tip (lateral view) devoid of cucullus. Spicular lamina mid-point not excessively widened, bearing central curved mid-line ridge along lamina. Dorsal spicular lamina smoothly and symmetrically curved.

*Female*: Ovary well developed, outstretched, extending to pharyngeal gland lobe, its distal end mostly reflexed, situated on right subventral side of mid-intestine. Oviduct straight and wide, with wrinkled surface. Spermatheca oval, situated ventrally and to left side of proximal part of oviduct, filled with ovoid cytoplasmic sperm 3 × 4 μm diam. Vagina 10–13 µm, sloping anteriorly to ventral body surface. Spermatheca opening from left side to pre-crustaformerial chamber via a spermathecal duct. Oviduct opening in pre-crustaformerial chamber from right side. Pre-crustaformerial chamber with small inner cavity, this chamber continuing proximally into crustaformeria. Pre-crustaformerial chamber and crustaformeria separated by sphincter. Crustaformeria formed by large cells containing cytoplasmic granules, joining with anterior uterus, walls consisting of large, flattened cells. Vagina 10–13 µm, sloping anteriorly to ventral body surface. Vulval anterior lip flap small, with apex directed anteriorly and long sides (lateral ridges) directed laterally. No vulval papillae or post-vulval surface transversal fold visible. Pair of three-celled structures situated laterally on both sides of vagina at uterus/post-uterine sac junction. Posterior vulval lip massive, expanded. Post-uterine sac (PUS) very wide, with sperm, posterior end hemispherical, not differentiated or with rudimentary ovary. Tail straight, hooked ventrally. Tail tip straight or slightly curved, conically rounded to digitate.

*Type habitat and locality*: Cultures on *Botryotinia fuckeliana*–potato dextrose agar medium (PDA) were started from individuals isolated from the phloem (0.5 cm deep) obtained from a dying ash, *Fraxinus mandshurica* (Oleaceae), showing symptoms of dieback, dark-colored ring in cross-section of wilted branches and trunk with galleries of larvae and pupae of *Hylesinus laticollis* Blandford (Curculionidae: Scolytinae). These were collected by Prof. Vyacheslav. B. Zvyagintsev in the Arboretum of the Far Eastern Research Institute of Forestry, on the bank of the Amur River (GIS 48.463611, 135.084444), Khabarovsk city, Khabarovsk Krai, Russia, 25 August 2018. Several cultures were started from dauer juveniles extracted from the dissected female adults of *H. laticollis* collected simultaneously from galleries in phloem.

*Etymology*: The specific epithet was given in honor of Prof. Vyacheslav B. Zvyagintsev (Belarus), who collected these valuable samples with dieback symptoms during his research survey of ash *Fraxinus* spp. dieback infections in plantations in Belarus and Russia.

*Type materials*: Type nematode material obtained from 2-week-old *Botryotinia fuckeliana* cultures. Holotype male (slide P-4522), 20 paratype females, 20 paratype males (slides P-4524-4528) deposited in the Nematode Collection of the Zoological Institute RAS, Saint Petersburg, Russia. Four paratype males and four paratype females also deposited in the Nematode Collection of Wageningen Agricultural University, The Netherlands, and four paratype males and four paratype females in the Nematode Collection of the University of California, Riverside, CA, USA.

*Diagnosis and relationships*: *Bursaphelenchus zvyagintsevi* sp. n. belongs to the *Abietinus* group [32] according to molecular phylogenetic analyses and morphological characters. The new species is characterized by body length of 470–676 μm, stylet = 12–15 μm, with flexible shaft, stylet base slightly and smoothly expanded into three ridges, but without knobs, lateral field with two incisures. Median bulb oval, pharyngeal dorsal lobe 4–6 body diam. long. Spermatheca oval filled with ovoid sperm 3–4 μm diam. Female PUS 53–83% of the vulva–anus distance and 3.6–5.8 vulval body diam. (VBD) long, filled with sperm, female tail straight, its tip conically rounded. Male with seven caudal papillae: 1 + 2 + 2 + 2, the mid-ventral unpaired P1 just anterior to the cloacal opening, the paired P2 level with the cloacal aperture, the paired P3 and paired pore-like ‘gland papilla’ P5 at the lateral edges of the very small conically rounded bursal flap. P4 pair absent.

Among the species of this group, the new species is similar to *B*. *sinensis* in body and spicule shapes. It differs from the latter in gland lobe 4–6 body diam. long vs. 2.1–4.3. Position of P5 (caudal gland-papillae of male) at 81 (74–86) of tail vs. 73 (67–77)%, rostrum well developed vs. rostrum not clearly differentiated, spicule capitulum straight vs. spicule capitulum rounded, condylus rounded, somewhat dorsally bent vs. condylus squared, somewhat ventrally bent in *B. sinensis* [33].

*Bursaphelenchus zvyagintsevi* sp. n. differs from *B. gerberae* in the following characters: spicule narrow with one line along lamina vs. spicule wide with numerous lines along lamina; rostrum rounded vs. rostrum sharply conical; a line drawn through the anterior-most points of the condylus and rostrum (along the capitulum) and extending the posterior dorsal lamina intersects ventrally at 17 (13–23)° vs. less than a 14° angle; lateral field with two vs. three incisures; bursa very small, conically rounded vs. bursa well developed, rounded; female tail straight vs. female tail strongly recurved ventrally [34].

*Bursaphelenchus zvyagintsevi* sp. n. differs from *B. aberrans* in the spicule shape: capitulum straight, condylus and rostrum well developed vs. condylus and rostrum fused, capitulum cap-like in *B. aberrans*; female tail straight vs. tail ventrally recurved [35].

The new species differs from *B. abietinus*, Braasch and Schmutzenhofer, 2000, in spicule devoid of cucullus vs. cucullus present in *B. abietinus*; PUS (post-vulval uterus sac) more than half of the vulva–anus distance vs. PUS less than 50% of the vulva–anus length; secretory-excretory pore at nerve ring or anterior vs. secretory-excretory pore posterior to nerve ring [36].

*Bursaphelenchus zvyagintsevi* sp. n. differs from *B. hellenicus*, Skarmotsos, Braasch and Michalopoulou, 1998, in narrow spicule without cucullus vs. wide spicule with small cucullus, lateral field with two lines vs. lateral field with three lines in *B. hellenicus* [37].

*Bursaphelenchus zvyagintsevi* sp. n. differs from *B. pityogeni*, Massey, 1974, in body length less than 700 µm vs. more than 800 µm in *B. pityogeni*; spicule with ventral velum and blunt rostrum vs. spicule without velum and sharp rostrum in *B. pityogeni* [38].

*Bursaphelenchus zvyagintsevi* sp. n. differs from *B. rainulfi*, Braasch and Burgermeister, 2002, in PUS more than half of the vulva-–anus distance vs. PUS less than 50% of the vulva-–anus length; spicule narrow with short condylus and one ridge along lamina vs. wide spicule with long prominent condylus and 2–3 ridges along lamina; female tail almost straight vs. female tail recurved [39].

*Bursaphelenchus zvyagintsevi* sp. n. differs from *B. varicauda*, Thong and Webster, 1983, in spicule narrow with short condylus vs. stout spicule with long prominent condylus; female tail tip without appendages vs. female tail tip with three finger-like processes in *B. varicauda* [40].

*Bursaphelenchus zvyagintsevi* sp. n. differs from *B. willi*, (Massey, 1974) Baujard, 1989, in PUS length = 4–6 body diam. long vs. 3 or less body diam. in *B. willi*; spicule narrow with short condylus vs. spicule very wide with long condylus, female tail tip conically rounded vs. female tail tip hemispherical, blunt in *B. willi* [38].

*Bursaphelenchus zvyagintsevi* sp. n. differs from *B. antoniae*, Penas, Metge, Mota and Valadas, 2006, in narrow spicule with ventral velum without cucullus vs. stout spicule with small cucullus and devoid of ventral velum; female tail 27–33 vs. 34–46 µm in *B. antoniae* [41,42].

*Bursaphelenchus zvyagintsevi* sp. n. differs from *B. chengi*, Li, Trinh, Waeyenberge, Moens, 2008, in body length of male 545 (470–676) vs. 723 (646–825) µm and female 566 (484–625) vs. 742 (661–828) µm; spicule narrow with short condylus and without cucullus vs. spicule very wide with long condylus and with distinct small cucullus [43].

*Bursaphelenchus zvyagintsevi* sp. n. differs from *B. niphades*, S. Tanaka, R. Tanaka, Akiba, Aikawa, Maehara, Takeuchi and Kanzaki, 2014, in vulva-–anus distance 123 (105–144) vs. 170 (143–207) µm; body length shorter than 750 µm vs. body longer than 750 µm; spicule narrow with short condylus, one ridge along lamina and without cucullus vs. spicule very wide with long condylus, 2–3 ridges along lamina and with distinct small cucullus [44].

*Bursaphelenchus zvyagintsevi* sp. n. differs from *B. parantoniae*, Maria, Fang, He, Gu and Li, 2015, in shorter tail c′ = 3.5 (3.1–4.0) vs. 4.7 (4.0–5.4); spicule narrow with ventral velum and devoid of cucullus vs. spicule stout without velum and with small cucullus in *B. parantoniae* [45].

*Bursaphelenchus zvyagintsevi* sp. n. differs from *B. sakishimanus*, Kanzaki, Okabe and Kobori, 2015, in body length of male 545 (470–676) vs. 930 (875–1036) µm; and female 566 (484–625) vs. 974 (929–1039) µm; shorter female tail: c′= 3.5 (3.1–4.0) vs. 6.8 (6.1–7.9); spicule narrow with ventral velum and one ridge along lamina, short round condylus and devoid of cucullus vs. spicule stout, without velum, two ridges along lamina, prominent conical condylus, and with small flat cucullus; lateral field with two incisures vs. three incisures in *B. sakishimanus* [46].

*Molecular characterization*: One D2-D3 of 28S and one ITS rRNA gene sequences were generated for this species. Phylogenetic position of *B. zvyagintsevi* sp. n. is given within some representatives of Aphelenchoidea (Figure 3) and the *Abietinus* group members (Figure 4) based on the analysis of D2-D3 of 28S and ITS rRNA gene sequences. Sequences of *B. zvyagintsevi* sp. n. clustered with that of *B. sinensis* in the D2–D3 28 rRNA gene tree (Figure 3 and Figure 4A) or with those of *B. sinensis* and *B. juglandis* (Figure 4B) in the ITS rRNA gene tree. The D2-D3 28 rRNA gene sequence of *B. zvyagintsevi* sp. n. differed from that of *B. sinensis* in 9.3% (66 bp). The ITS rRNA gene sequence of *B. zvyagintsevi* sp. n. differed from those of *B. sinensis* in 26.7–28.0% (212–223 bp) and from that of *B. juglandis* in 33.3% (285 bp).

### 2.2. Bursaphelenchus michalskii Tomalak and Filipiak, 2018

*Adults* (Figure 5, Table 1): Body very narrow and long, curved ventrally. Lateral field with three equidistant lateral incisures (two bands). Stylet base slightly expanded, but without distinct knobs. Cephalic annuli weakly visible under light microscopy, at least five annuli distinct. Lip region lobed, lateral lobes narrower than subventral and subdorsal ones. Median bulb pyriform, median valve located in posterior third of bulb, anterior third narrow, glandular, posterior two thirds spherical. Secretory–excretory pore located at nerve ring to one-half length of median bulb anterior to bulb.

*Male*: Similar to female in structure of anterior end. Testis situated on right subventral side of mid-intestine, long, anteriorly reflexed, zone of spermatids distinct, consisting of one or two quartets of large, separated cells, and a zone of granulated immature sperm cells located posterior to spermatids. Sperm gradually decreasing in size to ellipsoid mature sperm cells filling posterior 20% part of testis. Vas deferens with a wall of flat large polygonal cells, four cloacal lengths long. Tail strongly hooked, J-shaped, terminating with very small bursa 9–15 μm at borders, and 3-6.5 μm along midline. Bursal flap truncate with roundish concavity at center of distal end with two pointed tips on each lateral side. There are seven male tail papillae: mid-ventral unpaired P1 just anterior to cloacal opening, paired P2 almost at same level as P1 laterally (0–13% of tail length), paired P3 shifted to 41–58% of tail length from cloacal opening, and paired, small, pore-like P5 close to ventral mid-line at level of lateral edges of bursa 69 (66–75%). P5 pair may be considered as ‘gland papillae’ because of pore-like form, whereas other papillae are nipple-like. P4 pair absent. Spicule strong, wide, J-shaped, rostrum and condylus well developed and separated. Angle between lines along capitulum (condylus-rostrum) and extending spicule end = 13–23°, point of intersection ventral. Rostrum acute, sharp, offset from the spicule contour. Junction of rostrum and calomus rectangular. Condylus rounded, dorsally bent from contour of lamina. Capitulum of spicule with a distinct depression its depth one fifth of the capitulum length. Spicular tip with small flat cucullus. Spicular lamina mid-point widened with two or three central ridges along lamina. Dorsal spicular lamina smoothly and symmetrically curved.

*Female*: Ovary well developed, outstretched, extending to pharyngeal gland lobe, its distal end mostly reflexed, situated on right subventral side of mid-intestine. Oviduct straight and wide, with wrinkled surface. Spermatheca round to oval, situated ventrally and to left side of proximal part of oviduct, with oval granulated cytoplasmic sperm 7 × 5 μm. Spermatheca opening from left side to crustaformerial chamber via a spermathecal duct. Oviduct opening in pre-crustaformerial chamber from right side. Pre-crustaformerial chamber with small inner cavity, this chamber continuing proximally into crustaformeria. Pre-crustaformerial chamber and crustaformeria separated by sphincter. Crustaformeria formed by large cells containing cytoplasmic granules, joining with anterior uterus, walls consisting of large, flattened cells. Vagina 10-13 µm, cuticular, sloping anteriorly to ventral body surface, vulval flap absent. No vulval papillae or post-vulval fold visible. Pair of three-celled structures situated laterally on both sides of vagina at uterus/post-uterine sac junction. Posterior vulval lip massive, expanded. Post-vulval uterine sac (PUS) very wide, filled with sperm, its end hemispherical, not differentiated and devoid of rudimentary ovary. Tail straight. Tail tip straight, rounded to conically rounded, sometimes with a 2–3 µm mucron.

*Habitat and locality*: Cultures on *B. fuckeliana*–PDA medium were started from individuals isolated from the phloem (0.5 cm deep) obtained from a dying elm, *Ulmus minor* Miller, 1768 (Ulmaceae), showing symptoms of dieback, dark-colored ring in cross-section of wilted branches, trunk with galleries of larvae and pupae of *Scolytus jaroschewskii* Schevyrew, 1893 (Curculionidae: Scolytinae). These were collected by Dr. Alexander V. Petrov in the Samursky Forest Reserve, near the Caspian Sea (41.844639, 48.544833), Republic of Dagestan, Russia, 27 August 2021. Several cultures were started from dauer juveniles extracted from the dissected female adults of *S. jaroschewskii* collected simultaneously from galleries in phloem.

*Materials*: Material was obtained from 2-week-old fungus cultures. 20 females and 20 males (slides P-4529-4534) were deposited in the Nematode Collection of the Zoological Institute RAS, Saint Petersburg, Russia.

*Remarks*: There are some differences between the Dagestan population and Poland population. The spermatheca is not axial, but a side branching sac of the genital tube, sperm is ellipsoid and not spherical, and the median bulb is distinctly pyriform with posteriorly shifted inner valve vs. median bulb ovoid with central valve. Such new morphological features expand the characteristics of *B. michalskii*. The host plant and vector in the Dagestan population are also different: the vector *Scolytus jaroschewskii* and the host *Ulmus minor* for the new record vs. *S. scolytus* and *U. laevis* in the original description [47].

### 2.3. Characterization of Other Nematodes from Woody Plants

Molecular characterization of nematodes was carried out based on analysis of the D2–D3 of 28S rRNA gene. The phylogenetic positions of studied samples within closely related species and genera are given in Figure 3 and Appendix A.

#### 2.3.1. *Aphelenchoides*

*Aphelenchoides heidelbergi* (Appendix A)—Nematodes were extracted from galleries of bark beetles and maintained on *Botryotinia fuckeliana* culture. Sequences obtained from two nematode samples (Saint Petersburg and Dagestan) of this species were identical to each other and to that from Portugal. The percentage of its identity shared with other published sequences of this species from Australia, Turkey, the USA and China varied from 99.01 to 99.86%. The finding of *A. heidelbergi* is a new report of this species in Russia. This species was originally described from wood of the exotic pine, *Pinus radiata*, in Victoria, Australia, and then in the USA and Portugal [48,49,50].

Two unidentified *Aphelenchoides* sp.1 (Appendix A) and sp.2 were reported from *Quercus* and *Betula*, respectively. Sequences of *Aphelenchoides* sp.1 differed in 1.4% (10 bp) from each other and had the highest percentage of identity (86.6%) with *Aphelenchoides eldaricus.* The sequence of *Aphelenchoides* sp.2 was identical to that of unidentified *Aphelenchoides* sp. from Belgium (KX356787) (Figure 3).

*Aphelenchoides* is one of the largest genera under the order Aphelenchida and has the large number of hosts as well as a wide distribution [51]. According to the literature data, the following *Aphelenchoides* species were reported from living and freshly fallen wood in Russia: *A. clarus*, *A. hamatus*, *A. macromucrons*, *A. paramonovi. A. parasexlineatus*, *A. rhytium*, *A. saprophilus* and several other unidentified species [12,15,20,21,52,53].

#### 2.3.2. *Bursaphelenchus*

*Bursaphelenchus eremus* (Appendix A)—Nematodes were collected both from beetle galleries and beetles in oak groves, where most trees were dying. Its detection is a new report of this species in Russia. The sequence of this sample from Nizhny Novgorod clustered with that of *B. eremus* (AM396568) from Germany and differed from it by absence of an insertion (~19 bp) (Figure 3). The insertion was not observed in other *Bursaphelenchus* and aphelenchoid species and, perhaps, it is the result of a sequence reading mistake.

*Bursaphelenchus fraudulentus* (Appendix A)—Nematodes were extracted from galleries of the Cerambycidae beetle. Two sequences obtained from samples collected in Belarus and Moscow, Russia, were different in 1 bp (0.1%) and showed higher identity with sequences of this species deposited in the GenBank (Figure 3).

*Bursaphelenchus mucronatus* (Appendix A)—Samples were taken from galleries and living beetles *Polygraphus proximus*. The sequence of this sample collected in Buryatia showed the highest identity of 99.72% with sequences of this species from Germany (AM396572) and Mexico (EU295494) (Figure 3).

*Bursaphelenchus willibaldi* (Appendix A)—Specimens were extracted from wilted and dying *Quercus robur* trees, and bark with galleries of *Scolytus intricatus*. The sequence of this sample collected in Nizhny Novgorod was identical to that from Romania (MN879886) (Figure 3).

In this study, we described a new species, *B. zvuagintsevi* sp. n, and characterized several other representatives of this genus: *Bursaphelenchus eremus*, *B. fraudulentus*, *B. michalskii*, *B. mucronatus* and *B. willibaldi.* Reports of *B. eremus* and *B. michalskii* are new findings of these species for Russia. Thus, the total number of *Bursaphelenchus* species reported from Russia increased to 18 and included the following species: *B. africanus* [54], *B. borealis* [14,55], *B. crenati* [8], *B. eremus* (this study), *B. eroshenkii* [56], *B. fraudulentus* [21,55]; (this study), *B. fuchsi* [57], *B. hellenicus* [21,55], *B. hylobianus* [14,55], *B. kolymensis* [13,58], *B. leoni* [55], *B. michalskii* (this study), *B. mucronatus* [17,55,59,60] and this study, *B. paracorneolus* [55], *B. rockyi* [61], *B. ulmophilus* [6], *B. willibaldi* [28] and this study, and *B. zvuagintsevi* sp. n. (this study). Findings of *B. silvestris* [52,62] and *B. xylophilus* [55] require a confirmation.

#### 2.3.3. *Cryptaphelenchus*

*Cryptaphelenchus* sp.1 (Appendix A), sp.2 and sp.3 (Appendix A)—Specimens were obtained from beetle surface and galleries of beetles in the Tomsk region. Blastn search showed the highest identity for the sequences of all three species with the following *Cryptaphelenchus* in the GenBank: 95.74%—OP023328, *Cryptaphelenchus* sp., China, 92.73%—LR890130, *Cryptaphelenchus* sp., Italy, and 91.50%—KY385333, *C. baujardi*, respectively (Figure 3).

The genus *Cryptaphelenchus* contains more than 20 valid species. Most species were reported in association with frass and galleries of bark beetles, and are apparently mycetophagous. Several species of the genus *Cryptaphelenchus* were identified from wood in Russia: *C. aedili*, *C. borlossi*, *C. diversispicularis C. ipinius*, *C. koerneri* and *C. macrogaster* [10,11,14,19,52,63].

#### 2.3.4. *Deladenus*

*Deladenus posteroporus*—Nematodes were obtained from galleries of Scolytinae beetles. Two sequences of this species collected in the Magadan region are identical to each other and that of *D. posteroporus* described from packaging wood originating from Canada (KX094978) [64] (Appendix A). It is the first report of this species in Russia. The genus *Deladenus* contains more than 30 valid species [65,66]. A few species, including *D. laricis* [67], *D. obesus* [19], and several unidentified species [21] were reported in Russia.

#### 2.3.5. *Diplogasteroides*

*Diplogasteroides nix*—Nematodes were extracted from galleries of Scolytinae beetles. This sequence is identical to that of this species (LC145090) from Japan and very similar to those of *D. andrassyi* and *D. asiaticus* [68] (Appendix A). This is the first report in Russia and the second recorded outside the type locality in Japan. Another species, *Diplogasteroides sexdentati*, was reported in the former USSR and found in Russia [69].

#### 2.3.6. *Laimaphelenchus*

*Laimaphelenchus hyrcanus*—Specimens were extracted from galleries of the bark beetle *Scotylus* sp. The sequence of this sample showed the highest similarity (99.45 and 99.86%) with that of *L*. *hyrcanus* from Iran (KJ567061) and Serbia (KF881746), respectively (Figure 3). It is the first report of this species in Russia. This species was described from bark samples of cypress (*Cupressus* sp.) and jujube (*Ziziphus jujube*) from Iran [70] and bark of a wilting black pine (*Pinus nigra*) from Serbia [71,72].

*Laimaphelenchus* sp.1 (Appendix A)—Samples were collected from *A. platanoides* and a dying *Tilia cordata* tree infected with fungus *Kuehneromyces mutabilis*. These two sequences are similar (99.29%) to that of unidentified *Laimaphelenchus* sp. (OP012474) from China (Figure 3).

The genus *Laimaphelenchus* includes more than 15 valid species with worldwide distribution. Most species have been reported from soil, mosses, algae or tree bark [51]. Nine species of the genus *Laimaphelenchus (L. corticilis*, *L. deconincki*, *L. hyrcanus*, *L. montanus*, *L. pannocaudus*, *L. penardi*, *L. sapinus*, *L. silvaticus*, *L. ternarius)* and several unknown species were identified from woody plants in Russia [18,21,62,73,74,75] and this study.

#### 2.3.7. *Micoletzkya*

The genus *Micoletzkya* includes more than 20 nominal species and most of these species were isolated directly from the wood-boring Curculionidae including Scolytinae or from their breeding galleries. The association seems to be very specific and obligatory for the nematodes [76]. In this study, one unidentified *Micoletzkya* sp. was found as adult individuals and dauer juveniles collected on beetle surface in Buryatia, Baikal Nature Reserve. The sequence showed the highest similarity (98.87%) with that of *Micoletzkya* sp. (JX163966) from the bark beetle *Dryocoetes autographus* from Germany (Appendix A).

#### 2.3.8. *Parasitaphelenchus*

*Parasitaphelenchus* sp.1—Nematodes were extracted from galleries of *Ips amitinus* and from live beetles. The sequence showed the highest similarity (93.70%) with that of *Parasitaphelenchus*
*costati* (LC269967) from Japan (Figure 3). Currently, the genus *Parasitaphelenchus* contains 16 species. One species, *P. macrohami*, was described in Russia by Lazarevskaya [10]. The majority of the members were described in the early-middle 20th century. Therefore, many important typological characters are missing and no molecular information is available for these nominal species [77].

#### 2.3.9. *Parasitorhabditis*

*Parasitorhabditis* sp.1—The sequence of this sample collected in Buryatia showed the highest identity in 93.51% with a sequence of *Parasitorhabditis obtusa* (EF990724) from Germany (Appendix A). Nearly 44 species are currently known in the genus *Parasitorhabditis*. Some of the species are inquirenda due to inadequate descriptions [78]. The nematodes infect the digestive tract and occur in the lumen of the intestine of insects. Several species of *Parasitorhabditis* were reported in Russia, including *P. palliati*, *P. acanthocini*, *P. fuchsia* and *P. sexdentati* [19,52,62,79,80,81,82].

#### 2.3.10. *Tylenchid nematodes*

*Sphaerulariid nematode* sp.1, *tylenchid nematodes* sp.1 and sp.2—The sequences of these samples are clustered within Tylenchida and, based on phylogenetic analysis, they belong to unknown genera of the families Sphaerulariidae and Anguinidae, respectively (Appendix A).

### 2.4. General Discussion

Risk prediction at the early stages of biological invasions is the most cost-effective strategy to control invasive species [83]. To assess such risks, it is necessary to find refugia of potentially expandable species and identify associations both within a refugium and at migration points outside the native habitat.

The case of transformation into a true pathogen is well known for *Bursaphelenchus xylophilus*, because in its native areas in North and Central America, this species does not kill healthy pines. This invasive species invaded Asia in 1905 and Europe in 1999, and then transformed from a weak opportunistic pathogen to a forest pest of world quarantine importance for conifers, causing the PWD [84,85,86].

It is difficult to predict with high probability which opportunistic pathogenic species will become a true pest. Therefore, the most economical method is to collect data on distant associations of opportunistic phytopathogenic and other entomochore species associated with xylobiont insects. These data will make it possible to evaluate the ability of a harmful nematode to change vector and host plants, as well as to identify refugia and pathways of pathogen migration along trade routes.

The identification of entomochore saprobiotic and predatory nematodes in the associations of bark and longhorn beetles in refugia allows us to understand which nematode species can be used for biological control of plant pathogens during accidental anthropogenic transmission and during timber trade or the use of wood packaging. Data on entomochore nematode associations are of particular value for Russia. The fauna of xylobiont nematodes in this vast area are poorly studied. It can be assumed that there are many refugia of potentially dangerous species, as well as agents of biocontrol of these species. Russia has an active timber trade, which increases the risks of potentially dangerous invasive species migrating to purchasing countries in Europe, North America and Asia.

Newly recorded associations of nematodes, vectors and hosts of the *Bursaphelenchus* species contribute to understanding the origin and evolution and life cycles within the lineages (species groups) of the pathogenic genus *Bursaphelenchus* and may be used for the mapping of possible refugia of the species.

*Abietinus* group: *Bursaphelenchus zvyagintsevi* sp. n. is most similar to *B. sinensis* and *B. abberans*, both in specular and general morphology and molecular characteristics. However, the data on the associations of the latter two species are not complete, because their vectors were not identified. Moreover, *B. abberans* is suspected to be conspecific with of *B. sinesis* [35,87]. Both species were recorded only from wood of *Pinus* spp. trees. The species list of pine hosts for *B. sinensis* is as follows: *Pinus densifora*, *P. thunbergii* and *Pinus* sp. found in China, Japan and Korea [33,88,89] and all native records were from East Asia. The first detection of *B. sinensis* was on wood packaging material in Austria. It was intercepted by quarantine services, but other secondary dispersal cases to Europe are not excluded. *Bursaphelenchus aberrans* has been reported from *Pinus massoniana* in China [35,90] and *Pinus merkusi* in Thailand [91];. In Europe it was intercepted by quarantine services in Austria in load boards and pallets, tree not specified [92]. The present record of *B. zvyagintsevi* sp. n. with exact vector data and the deciduous tree host *Fraxinus* gives new information on the *B. sinensis* clade within the *Abietinus* group. However, it cannot be interpreted phylogenetically because of a lack of data for vectors of *B. aberrans* and *B. sinensis*. The origin from native refugia in East Asia and the tree host shift from coniferous to deciduous hosts are evident for *B. zvyagintsevi* sp. n.

*Eggersi* group: The new Caucasian record of *B. michalskii* is the second-furthest distance record of this rare species that was originally found in the Malta urban forest, Poznan, Poland [47]. The host plant and vector in our population are different but phylogenetically very close: the vector *Scolytus jaroschewskii* and the host *Ulmus minor* is a new record vs. *S. scolytus* and *U. laevis* in the original description [47]. Based on the new Dagestan record, the conclusion of Kanzaki et al. [93,94] is confirmed: the subgroup 2 of the *Eggersi* group is associated with the *Scolytus* spp. vectors and the Ulmaceae woody plant hosts. The *B. michalskii* species in far-distanced natural refugia is associated with the *Scolytus* spp. vectors and elm hosts of the family Ulmaceae. It is highly likely that coevolution within nematode–vector–host association took place.

Another species within the *Eggersi* group is *B. eremus*, which was recorded in the Nizhni Novgorod region, in galleries of *Scotylus intricatus* in the wilted *Quercus robur*. All records of this nematode are known from the same vector species and *Quercus* spp. [95,96,97].

*Fungivorus* group: The *B. willibaldi* extracted from in *Quercus robur* bark and *Scolytus intricatus* vector is the first distinct identification of the vector species for this nematode. The original description of *B. willibaldi* from Germany [98] was based the *Pinus sylvestris* chips; other records are of its associations in *Abies* sp., Romania [99], and in *Fagus* sp., Iran [100]. It may be concluded that *B. willibaldi* is presumably specific for the Pinaceae and Fagaceae host plants, and for *Scolytus* spp. vectors. Generally, the *Fungivorus* group is neither specific to hosts or vectors, nor distinctly pathogenic; these species are weak opportunistic pests participating in dead organic matter destruction [9].

*Xylophilus* group: The records of *B. mucronatus* in Pinaceae hosts and *B. fraudulentus* in deciduous hosts only confirmed the known host specificity and distribution areas of these species [101].

Among other taxa, the insect endoparasites *Deladenus posteroporus*, *Parasitaphelenchus* sp.1, and *Parasitorhabditis* sp.1 are possible biocontrol agents for insect vectors, especially *Parasitorhabditis* sp.1 for the invasive bark beetle *Polygraphus proximus* and *Parasitaphelenchus* sp.1 for *Ips* sp. The predator *Micoletzkya* sp.1 presumably may be used to control *Bursaphelenchus* spp. pending development of their cultivation. Species of the *Aphelenchoides*, *Laimaphelenchus* and *Cryptaphelenchus* are the fungal feeders associated with insect vectors and involved in the dead wood destruction; presumably, they restrict the multiplication of the pathogenic fungi in native habitats.

## 3. Conclusions

The identification of natural refugia of potential plant pathogens, which include opportunistic pathogenic species of the genus *Bursaphelenchus* and other entomochore Aphelenchoididae, is necessary for risk assessment and the prevention of new biological invasions. In these natural habitats of the *Bursaphelenchus* spp., entomoparasitic nematodes may be potential biocontrol agents of their insect vectors, and mycophagous nematodes from the vector galleries presumably prevent the intense expansion of entomochore phytopathogenic fungi vectored by insects together with nematodes. All of these nematode species were identified, either as pathogens or as potential biocontrol agents. Simultaneously, the mapping of xylobiont nematode associations was carried out to understand directions of possible expansion pathways. An integrated approach was used involving both molecular and morphological identification. During the present study, the refugia of rare species of the genera *Bursaphelenchus*, *Aphelenchoides*, *Laimaphelenchus*, *Parasitaphelenchus*, *Deladenus*, *Diplogasteroides*, *Parasitorhabditis* and others were identified. Their genetic characterization and the detailed morphological description of two rare *Bursaphelenchus* species, *B. zvyagintsevi* sp. n. and *B. michalskii*, are given. It is revealed that the species of the *Bursaphelenchus* groups *Abietinus*, *Eggersi* and *Xylophilus* are presumably characterized by narrow vector and host plant specificity, restricted mostly at the genus level of the associates.

## 4. Materials and Methods

### 4.1. Sample Collection and Processing

Wood and bark samples were collected during the surveys conducted in different regions of Russia and Belarus in 2014–2022 (Table 1). The 150–500 cm^3^ bark and wood samples with galleries of insects were obtained from trunks and 5–10 cm diam. branches using a folding garden saw, field axe, pruner, and slotted screwdriver with sharpened 3–5 mm tip.

Nematodes were extracted from wood and bark samples for 6–16 h at room temperature using a modified Baermann funnel technique [102]. The bark galleries in the phloem and outer layer of wood were first cut with a scalpel and then put in an extraction chamber with a double cotton filter (6 cm diam. cotton pad and 10 × 10 cm cotton-cellulose tissue). This chamber was put into a plastic-bottomed glass into which tap water was poured, thus, the sample was covered with water rising from the bottom of the glass through a filter. Actively moving nematodes passed down through a filter, while the debris particles were trapped. It was determined that before sorting into species and processing, isolated nematode suspensions can be stored for up to 3–4 months at +8 °C in 0.5 mL tubes with a drop of 0.15–0.25 mL of tap water or Ringer’s solution to ensure adequate oxygen access.

Dauer juveniles from bark and longhorn beetles collected from bark galleries and elytra of insects were submerged in Ringer’s solution (0.9% NaCl) for 30 min. Additionally, insects and their senior larvae were cut in a drop of Ringer’s solution on a plastic Petri dish with a scalpel. The suspensions extracted from bark and obtained from insects were studied under the stereomicroscope Mikromed MC-5 Zoom LED and the nematodes were picked out with a needle to select different taxa for propagation in vitro. Extracted nematodes were cultivated on laboratory cultures of the fungus *Botryotinia fuckeliana*, (de Bary) Whetzel, 1945 (=*Botrytis cinerea* Pers., 1794), growing on 2% potato dextrose agar medium at 22 °C [28,103]. *Aphelenchoides heidelbergi*, *Aphelenchoides* sp.1, *Bursaphelenchus eremus*, *B. fraudulentus*, *B. michalskii*, *B. mucronatus*, *B. willibaldi* and *B. zvyagintsevi* sp. n. were maintained in the fungus culture and used for morphological and molecular studies.

### 4.2. Morphological Study

For the morphological study, specimens were fixed in hot TAF (4% formaldehyde with addition of 2 mL of triethanolamine for 100 mL of solution) according to the technique developed by Ryss [102,104]. Fixed nematodes were processed in glycerin according to Ryss [105] and embedded in permanent slides. Statistical morphometric parameters were determined in MS Excel. All slide-mounted nematodes were measured and photographed under an automated Leica DM5000 B microscope with differential interference contrast (DIC) and a Leica DFC320 (R2) digital camera with Leica DFC Twain Software for PC and Leica IM50 Image Manager for PC. Illustrations were made using a camera lucida and series of photographs. All measurements were analyzed using the software ImageJ 1.48v (http://imagej.nih.gov/ij accessed on 10 January 2023). Permanent slides were deposited in the Nematode Collection of ZIN RAS, Saint Petersburg.

### 4.3. Molecular and Phylogenetic Study

DNA was extracted from several specimens using the proteinase K protocol. DNA extraction and PCR protocols were performed according to Subbotin [106]. The following primer sets were used in this study: (i) D2A (5′—ACA AGT ACC GTG AGG GAA AGT TG—3′) and D3B (5′—TCG GAA GGA ACC AGC TAC TA—3′) amplifying the D2–D3 expansion segments of the 28S rRNA gene, and (ii) TW81 (5′—GTT TCC GTA GGT GAA CCT GC—3′) and AB28 (5′—ATA TGC TTA AGT TCA GCG GGT—3′) amplifying the ITS rRNA gene. The successfully amplified fragments were sequenced using the primer pairs used in PCR. The new sequences for each gene were edited, compared with other sequences available in GenBank database, and the relevant sequences were retrieved. New sequences were aligned using ClustalX 1.83 (parameters: gap opening 5.0 and gap extension—3.0) with corresponding selected and published gene sequences of nematodes [9,64,68,76,107]. Sequence datasets were analyzed with Bayesian inference (BI) using MrBayes 3.1.2 and PAUP 4.0 as described by Subbotin [108]. The new sequences were submitted to the GenBank database under accession numbers indicated in Table 1 and phylogenetic trees.

Species delimitation was performed using an integrated approach that considered morphological and morphometric evaluation combined with molecular criteria based on phylogenetic inference and sequence analyses.

## Figures and Tables

**Figure 1 plants-12-00382-f001:**
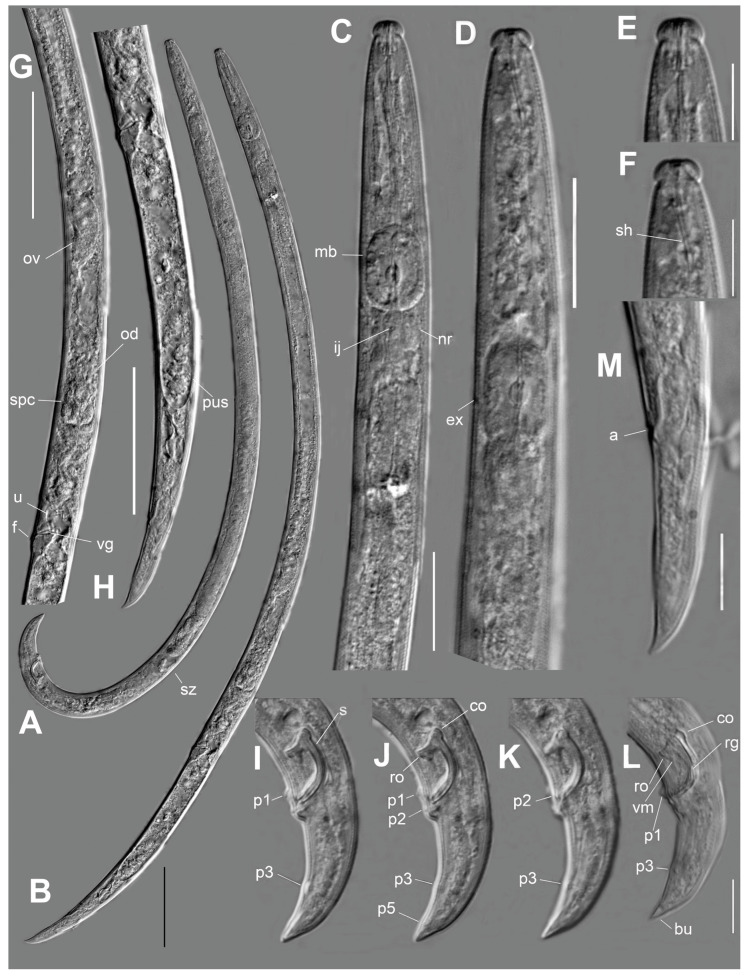
*Bursaphelenchus zvyagintsevi* sp. n. (**A**)—male. (**B**)—female. (**C**)—female anterior end. (**D**)—male anterior end. (**E**)—female lip region. (**F**)—male lip region. (**G**)—female anterior genital tube. (**H**)—post-vulval part of female. (**I**–**L**)—caudal part of male. (**M**)—tail of female. Abbreviations: a—anus, bu—bursal flap, co—condylus, ex—secretory-excretory pore, f—vulval flap, ij—oesophago-intestinal junction, mb—median bulb, nr—nerve ring, od—oviduct, ov—ovary, P1, P2, P3, P5—male caudal papilla, pus—post-vulval uterus sac, rg—ridge along specular lamina, ro—rostrum, s—spicule, sh—stylet shaft (flexible), spc—spermatheca, sz—zone of spermatids, u—uterus, vg—vagina, vm—ventral velum of spicule. Scale—50 µm for (**A**,**B**,**G**,**H**); 20 µm for (**C**,**D**); 10 µm for rest.

**Figure 2 plants-12-00382-f002:**
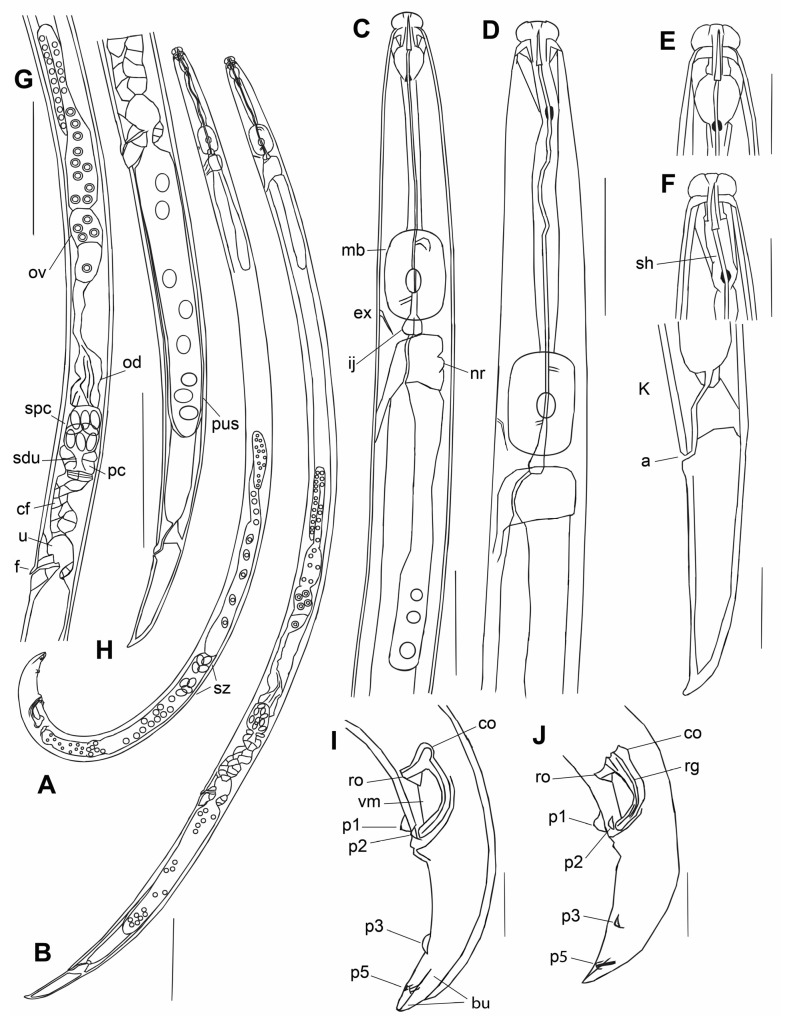
*Bursaphelenchus zvyagintsevi* sp. n. (**A**)—male. (**B**)—female. (**C**)—female anterior end. (**D**)—male anterior end. (**E**)—female lip region. (**F**)—male lip region. (**G**)—female anterior genital tube. (**H**)—post-vulval part of female. (**I**,**J**)—caudal part of male. (**K**)—tail of female. Abbreviations: bu—bursa, cf—crustaformeria, pc—pre-crustaformerial chamber, sdu—duct of spermatheca to pre-crustaformerial chamber, other abbreviations as in Figure 1. Scale—50 µm for (**A**,**B**,**G**,**H**); 20 µm for (**C**,**D**); 10 µm for rest.

**Figure 3 plants-12-00382-f003:**
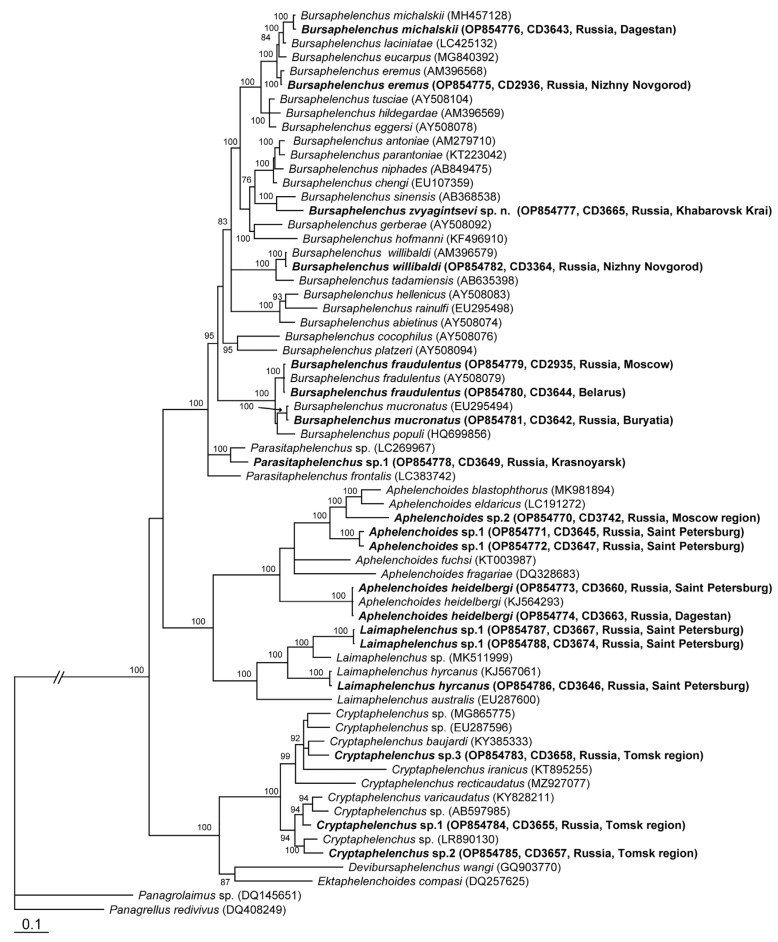
Phylogenetic position of the studied species relationships within some representatives from the superfamily Aphelenchoidea as inferred from Bayesian analysis of the D2–D3 of 28S rRNA gene sequences (ntax = 65 nchar = 913). Posterior probability values more than 70% are given on appropriate clades. New sequences are indicated in bold.

**Figure 4 plants-12-00382-f004:**
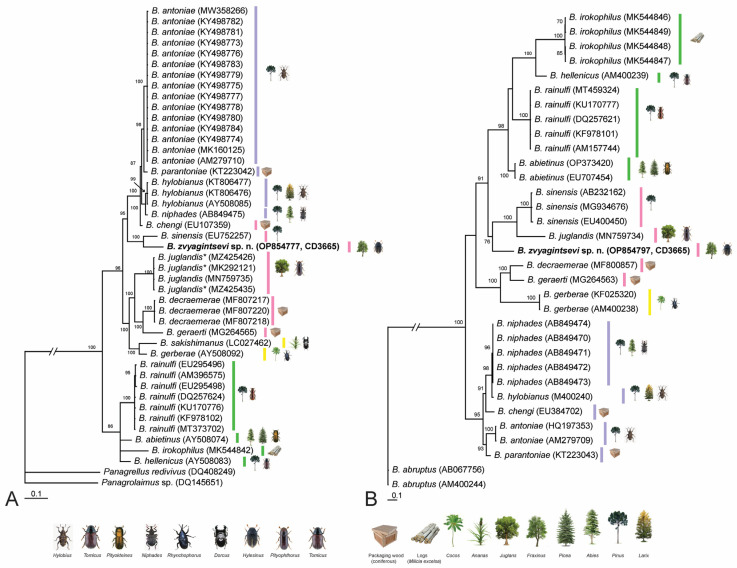
Phylogenetic relationships of *Bursaphelenchus zvyagintsevi* sp. n. within some species of the *Abietinus* group *sensu*, Ryss et al. [32], as inferred from Bayesian analysis of the D2–D3 of 28S rRNA (ntax = 45 nchar = 820) (**A**) and ITS rRNA (ntax = 33 nchar = 1356) (**B**) gene sequences. Posterior probability values more than 70% are given on appropriate clades. New sequence is indicated in bold. *—identified as *Bursaphelenchus* sp. in the GenBank.

**Figure 5 plants-12-00382-f005:**
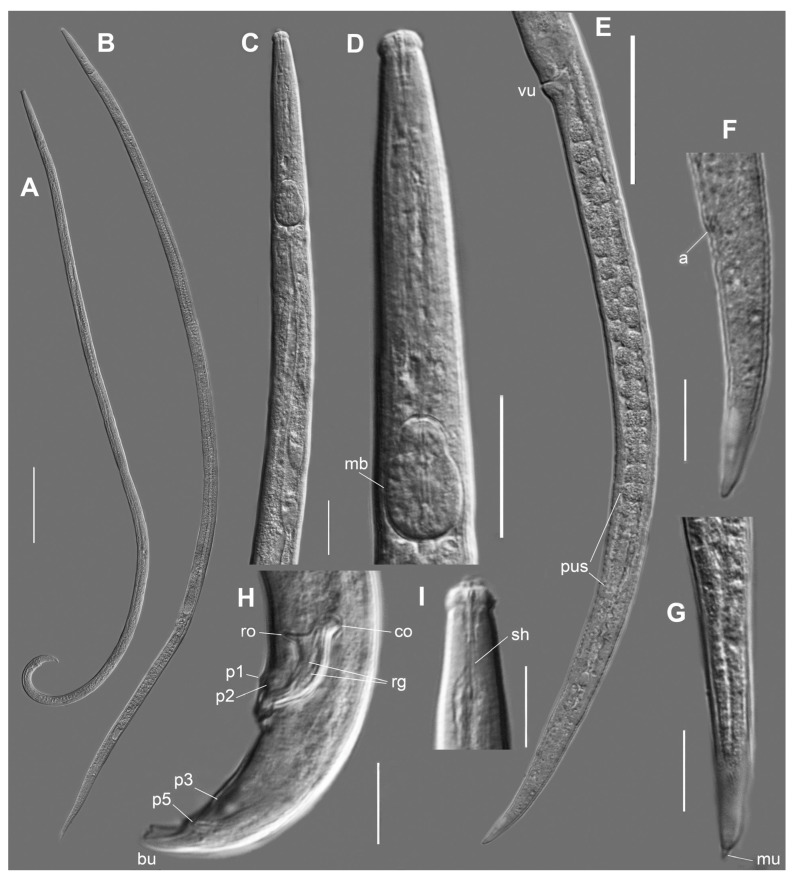
*Bursaphelenchus michalskii*, Dagestan. (**A**)—male. (**B**)—female. (**C**,**D**)—female anterior end. (**E**)—posterior body end of female. (**F**,**G**)—female tails. (**H**)—caudal part of male. (**I**)—male lip region. Abbreviations: mu—mucron, vu—vulva, other abbreviations as in Figure 1. Scale—100 µm for (**A**,**B**); 20 µm for (**C**,**D**); 50 µm for (**E**); 10 µm for rest.

**Table 1 plants-12-00382-t001:** Species and populations of nematodes from Russia and Belarus characterized in the present study.

Species	Sample Code	Location	GPS Coordinates	Associated Plant/Insect	GenBank Accession Number
*Aphelenchoides heidelbergi*	CD366	Saint Petersburg, Bol’shoy Prospekt Vasil’yevskogo Ostrova 25	59.939251; 30.281787	*Quercus robur*/*Scolytus* sp.	OP854773
*A. heidelbergi*	CD3663	Republic of Dagestan, Samurskiy Les	41.844639; 48.544833	*Elaeagnus latifolia*/*Scolytus jaroschewskii*	OP854774
*Aphelenchoides* sp.1	CD3645	Saint Petersburg, Bol’shoy Prospekt Vasil’yevskogo Ostrova	59.939251; 30.281787	*Q. robur*/*Scolytus* sp.	OP854771
*Aphelenchoides* sp.1	CD3647	Saint Petersburg, Isaakiyevskiy Skver	59.932579; 0.307305	*Q. robur*/*Scolytus* sp.	OP854772
*Aphelenchoides* sp.2	CD3742	Nizhny Novgorod	-	*Betula* sp./-	OP854770
*Bursaphelenchus* *eremus*	CD2936	Nizhny Novgorod, Strigino	56.198483; 43.761133	*Q. robur*/*Scolytus intricatus*; *Plagiontus* sp.	OP854775
*B. fraudulentus*	CD2935	Moscow, Main Botanical Garden of the Russian Academy of Sciences	55.841033; 37.607117	*Q. robur*/Cerambycidae	OP854779
*B. fraudulentus*	CD3644	Belarus, Gomel Region	52.478967; 29.419450	*Q. robur*/Cerambycidae	OP854780
*B. michalskii*	CD3643	Republic of Dagestan, Samurskiy Les	41.844639; 48.544833	*Ulmus minor*/*S. jaroschewskii*	OP854776
*B. mucronatus*	CD3642	Buryatia, Baikal Nature Reserve	51.415833; 104.821794	*Abies sibirica*/*Monochamus galloprovincialis*	OP854781
*B. willibaldi*	CD3664	Nizhny Novgorod, Botanical Garden of Lobachevsky University	56.254267; 44.006583	*Q. robur*/*S. intricatus*	OP854782
*B. zvyagintsevi* sp. n.	CD3665	Khabarovsk Krai, Arboretum of the Far Eastern Research Institute of Forestry	48.463611; 135.084444	*Fraxinus mandshurica*/*Hylesinus laticollis*	OP854777
*Cryptaphelenchus* sp.1	CD3655	Tomsk region, Luchanovo- Ipatovsky Settlement Cedar Forest	56.333228; 85.050120	*Pinus sibirica*/*Ips sexdentatus*	OP854784
*Cryptaphelenchus* sp.2	CD3657	Tomsk region, Malinovka	56.696938; 85.362820	*A. sibirica*/*Polygraphus proximus*	OP854785
*Cryptaphelenchus* sp.3	CD3658	Tomsk region, Zavarzino	56.472507; 85.092819	*A. sibirica*/*P. proximus*	OP854783
*Deladenus posteroporus*	CD3652	Magadan region, Omsukchansky District	61.180106; 153.900808	*Larix sibirica*/Scolytinae	OP854789
*Deladenus posteroporus*	CD3668	Magadan region, Omsukchansky District	61.180106; 153.900808	*L. sibirica*/Scolytinae	OP854790
*Diplogasteroides nix*	CD3673	Magadan region, Omsukchansky District	61.180106; 153.900808	*L. sibirica*/Scolytinae	OP854796
*Laimaphelenchus hyrcanus*	CD3646	Saint Petersburg, Isaakiyevskiy Skver	59.932579; 30.307305	*Q. robur*/*Scolytus* sp.	OP854786
*Laimaphelenchus* sp.1	CD3667	Saint Petersburg, Russian Museum, Mikhailovsky Garden	59.940949; 30.332522	*A. platanoides*/Scolytinae	OP854787
*Laimaphelenchus* sp.1	CD3674	Saint Petersburg, Russian Museum, Mikhailovsky Garden	59.939623; 30.329235	*Tilia cordata*/*Scolytus* sp.	OP854788
*Micoletzkya* sp.1	CD3672	Buryatia, Baikal Nature Reserve	51.563500; 105.394778	*A. sibirica*/*P. proximus*	OP854795
*Parasitaphelenchus* sp.1	CD3649	Krasnoyarsk	56.103451; 92.959464	*A. sibirica*/*Ips* sp.	OP854778
*Parasitorhabditis* sp.1	CD3671	Buryatia, Baikal Nature Reserve	51.563500; 105.394778	*A. sibirica*/*P. proximus*	OP854794
*Sphaerulariid nematode*	CD3653	Saint Petersburg, Ekaterinhof Park	59.902938; 30.267495	*Lindera angustifolia*/Scolytinae	OP854791
*Tylenchid nematode* sp.1	CD3666	Saint Petersburg, Russian Museum, Mikhailovsky Garden	59.940617; 30.332185	*T. cordata*/*Scolytus* sp.	OP854793
*Tylenchid nematode* sp.2	CD3648	Saint Petersburg, Russian Museum, Mikhailovsky Garden	59.939513; 30.328864	*U. laevis*/*S. multistriatus*	OP854792

**Table 2 plants-12-00382-t002:** Measurements (in μm) and indices of *Bursaphelenchus zvyagintsevi* sp. n and *B. michalskii*. All measurements are in μm and in the form: mean ± s.d. (range).

	Species	*Bursaphelenchus zvyagintsevi* sp. n.			*Bursaphelenchus michalskii* Dagestan Population	
Character		Holotype Male	Male	Female	Male	Female
n		1	20	20	20	20
L		562	545 ± 65 (470–676)	566 ± 47 (484–625)	901 ± 119 (699–1052)	1035 ± 164 (817–1277)
a		34.1	35.2 ± 2.4 (33.1–40.2)	32.1 ± 2.6 (28.0–36.7)	64.4 ± 7.0 (53.8–73.7)	56.3 ± 6.3 (50.2–65.5)
b		9.9	9.4 ± 1.5 (8.3–12.8)	9.0 ± 0.9 (7.9–10.3)	13.9 ± 1.2 (11.7–15.4)	15.5 ± 1.7 (13.0–18.2)
b’		4.7	4.3 ± 0.7 (3.8–5.7)	4.3 ± 0.4 (3.7–4.7)	5.9 ± 0.6 (5.0–6.9)	6.3 ± 0.7 (5.4–7.3)
c		17.6	19.1 ± 2.1 (17.1–22.5)	18.7 ± 1.2 (17.3–20.4)	31.1 ± 3.3 (28.0–36.3)	23.7 ± 2.2 (19.2–25.5)
c’		2.3	2.0 ± 0.2 (1.7–2.3)	3.5 ± 0.4 (3.1–4.0)	2.3 ± 0.2 (2.1–2.6)	4.6 ± 0.4 (4.1–5.1)
V (%)		-	-	73 ± 2 (70–76)	-	71.4 ± 0.9 (70.1–72.7)
Stylet		14	13.7 ± 1.1 (12–15)	13.6 ± 0.7 (12–15)	13 ± 1 (11–14)	14.0 ± 1.3 (12.0–15.5)
Stylet cone/stylet (%)		43	47 ± 5 (39–54)	49 ± 4 (43–56)	50 ± 4 (45–56)	52 ± 5 (45–59)
Cephalic region diam.		6.5	5.5 ± 0.9 (4.0–6.5)	6.1 ± 0.6 (5.0–7.0)	6 ± 0.5 (5–6.5)	6.5 ± 0.5 (6–7)
Cephalic region height		3.0	2.8 ± 0.4 (2.5–3.5)	2.9 ± 0.5 (2.0–3.5)	3.1 ± 0.3 (2.5–3.5)	3.3 ± 0.3 (3–3.5)
Median bulb length (L)		16	14.8 ± 0.9 (13–16)	16.6 ± 1.4 (15.0–19.0)	14.3 ± 1.4 (13.0–16.5)	14 ± 2.9 (11–18)
Median bulb diam. (D)		10	9.8 ± 0.7 (9–11)	11.0 ± 0.9 (10.0–12.0)	8.5 ± 1.5 (6.0–10.0)	9.4 ± 1.0 (8–11)
Median bulb L/D		1.6	1.5 ± 0.2 (1.2–1.7)	1.5 ± 0.1 (1.4–1.7)	1.7 ± 0.3 (1.4–2.3)	1.5 ± 0.2 (1.2–1.7)
Excretory pore from anterior		58	66 ± 5 (58–71)	73 ± 5 (65–80)	71 ± 10 (57–84)	80 ± 7 (71–89)
Nerve ring posterior border from anterior		78	74 ± 4 (70–78)	78 ± 3 (72–83)	81 ± 7 (74–92)	85 ± 9 (74–99)
Pharynx		57	58.3 ± 3.4 (53–62)	63 ± 4 (57–71)	65 ± 5 (57–73)	67 ± 6 (59–76)
Anterior to gland lobe end		120	127.9 ± 7.7 (119–137)	131 ± 4 (124–138)	154 ± 14 (135–173)	163 ± 22 (141–192)
Gland lobe		63	69.6 ± 4.7 (63–76)	68 ± 6 (57–75)	89 ± 11 (75–103)	97 ± 16 (81–118)
Gland lobe/body diam. at median bulb		4.3	5.1 ± 0.6 (4.1–5.8)	4.7 ± 0.6 (3.8–5.4)	7.1 ± 0.8 (5.9–8.8)	7.1 ± 0.9 (6.0–8.4)
Body diameter (BW)		16.5	15 ± 1 (13–17)	18 ± 2 (15–21)	14.1 ± 1.9 (12.0–16.5)	18.4 ± 2.5 (13.5–21)
PUS		-	-	82 ± 10 (62–98)	-	179 ± 20 (140–200)
Vulva–anus distance		-	-	123 ± 14 (105–144)	-	251 ± 36 (208–298)
Vagina		-	-	11 ± 1 (10–13)	-	11.6 ± 1.3 (9.5–12.5)
PUS/BW at vulva		-	-	4.7 ± 0.8 (3.6–5.8)	-	9.8 ± 0.6 (9.1–10.6)
PUS/vulva–anus distance (%)		-	-	67 ± 9 (53–83)	-	72 ± 7 (63–81)
Tail		32	29 ± 3 (25–32)	30 ± 2 (27–33)	29 ± 3 (24–34)	44 ± 5 (36–50)
Tail diam. at anus		14	14 ± 1 (13–16)	9 ± 1 (8–10)	13 ± 1 (12–15)	9 ± 1 (8.0–10.5)
Annuli (width of 10 at mid-body)		8.0	8.0+0.9 (7.0–9.5)	8.8 ± 0.8 (7.5–10.0)	10.9+2.4 (8.0–14.0)	11.1+2.0 (8.0–14.5)
Spicule length (midline arc)		16.5	16.5 ± 0.8 (15–17.5)	-	13.7 ± 1.3 (12.4–15.5)	-
Spicule length (dorsal lamina arc)		19	19.1 ± 0.7 (18–20)	-	17.6 ± 1.0 (16–19)	-
Spicule length (ventral lamina arc)		17	15.3 ± 1.4 (13.7–17)	-	12.2 ± 1.9 (10–15)	-
Distance between ends of condylus and rostrum		6.5	6.5 ± 0.8 (5–7.5)	-	7.7 ± 0.9 (6.3–8.8)	-
Spicule width posterior to rostrum (lateral view)		2.5	2.6 ± 0.3 (2–3)	-	3.5 ± 0.4 (2.8–4.2)	-
Angle between lines: along capitulum (condylus–rostrum) and extending spicule end (dorsal intersection)		13	17.4 ± 3.1 (13–23)°	-	6.0 ± 2.2 (3–9)°	-

## Data Availability

Sequencing data are available from the NCBI database, accession # OP854770-OP854797. Nematode materials were deposited in the Nematode Collection of the Zoological Institute of the Russian Academy of Sciences (UFK ZIN RAS). All other relevant data are included within the manuscript and Appendix A.

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
