# Peer review of "New Records of Wood- and Bark-Inhabiting Nematodes from Woody Plants with a Description of Bursaphelenchus zvyagintsevi sp. n. (Aphelenchoididae: Parasitaphelenchinae) from Russia"

_plants, 2023, doi:10.3390/plants12020382_

Round 1

Reviewer 1 Report

Line 16-17: Combine the two sentences as it will add some context to the “great economic importance” of the nematodes being considered. Similarly, the information between lines 17 and 19 can be combined in one sentence as these are related.

Line 20: Add a full stop before “Using”

Line 37-38: These two lines are repeated (appeared in abstract as well), modify the sentences either in the abstract or in introduction

Line 39: Add some numbers (could be in Russia or global impact) supporting the economic importance of nematodes under study

Line 42: Re-write for clarity

Add reference(s) for info provided in lines 49-51

Line 48: Replace “all” with “the”

Line 50: Add space between “bacteria” and “are”

Line 51: Add a comma after “study”

Line 52: Replace “in systems” with “from systems”?

Line 55: Add a comma after “ecosystems”

Line 149: Remove an extra “end”

Table 2: The heading “character” and “species” appeared in the same column

Line 582: Replace the heading title “Nematode samples” with “Sample collection and processing” to address everything included under that heading

Line 603: Remove “of”

Line 611-612: Change “the morphological study” with “morphological analysis”                                                                                                                                       

Line 626: Add “done” before “according”

Author Response

We are grateful for all your comments and suggestions.
Line 16-17: Combine the two sentences as it will add some context to the “great economic importance” of the nematodes being considered. Similarly, the information between lines 17 and 19 can be combined in one sentence as these are related.
Corrected
Line 20: Add a full stop before “Using”
Corrected
Line 37-38: These two lines are repeated (appeared in abstract as well), modify the sentences either in the abstract or in introduction
Corrected
Line 39: Add some numbers (could be in Russia or global impact) supporting the economic importance of nematodes under study
Unfortunately, there are not reliable numbers and we tried to avoid to make such modification in the sentences. 
Line 42: Re-write for clarity
Corrected
Line 48: Replace “all” with “the”
Corrected
Line 50: Add space between “bacteria” and “are”
Corrected
Line 51: Add a comma after “study”
Corrected
Line 52: Replace “in systems” with “from systems”?
Corrected
Line 55: Add a comma after “ecosystems”
Corrected
Line 149: Remove an extra “end”
Corrected

Table 2: The heading “character” and “species” appeared in the same column
Corrected
Line 582: Replace the heading title “Nematode samples” with “Sample collection and processing” to address everything included under that heading
Corrected
Line 603: Remove “of”
Corrected
Line 611-612: Change “the morphological study” with “morphological analysis
We prefer to keep “study”.
Line 626: Add “done” before “according”
Corrected

Thanks!
Sincerely
Sergei Subbotin and Alexander Ryss

Reviewer 2 Report

Dear Alexander and Sergei

With great interest I read your article entitled “New records of wood- and bark-inhabiting nematodes from woody plants with a description of Bursaphelenchus zvyagintsevi sp. n. (Aphelenchoididae: Parasitaphelenchinae) from Russia” submitted to plants.

Within your article, you describe findings of wood- and bark-inhabiting nematodes from various collections in Russia and Belarus over a period of 8 years. Nematodes were in most cases isolated as dauer juveniles from insect galleries or vector insects. For species identification, you used both, morphological and molecular features. In addition to various novel descriptions of species for Russia, you describe a novel species Bursaphelenchus zvyagintsevi sp. n. (Aphelenchoididae: Parasitaphelenchinae) in honor of Prof. Zvyagintsev and draw comparisons between the recently identified B. michalskii from Dagestan with the original Poland population.

The manuscript is well written and follows the rules of species descriptions. The descriptions of all morphological features are scientifically sound. The manuscript is of interest to forest nematologists and forest ecologists and should be published after some revisions:

I have some minor remarks: It´s not clear, why the manuscript contains a detailed description of B. michalskii. A rationale should be provided in the introduction. In addition, the manuscript is lacking a perspective, how species diversity can be captured nowadays. There are NGS approaches published that provide e.g. insights into nematode diversity in deadwood. Hence, it should be discussed why many of your isolated individuals are not determined/ identified to the species level. Wat are the challenges?

Unfortunately, the manuscript contains many typos and needs careful re-reading. I may provide some examples:

Abstract ln 20: elytrae of insects.

It´s B. fraudulentus not fradulentus (this is misspelled in the entire manuscript)

Ln 31: refugia instead of refuges

Ln 45: collected in Russia and Belarus!

Lns 51-53: I don´t understand this sentence, rephrase.

Ln 160: I would move Blandford to line 156 (first mentioning of the species)

Lns 156ff: the entire paragraph contains typos, e.g. it´s one sequence and not sequences…

Ln 322: Ulmaceae

Ln 356: ITS

Ln 450: I would write wood-boring Curculionidae including Scolytinae or only write Scolytinae

Ln 452: bark and dauers? -> dauer juveniles

Last: Sequences not available yet, I checked at GenBank. Please make sure they are accessible upon publication.

I wish you all the best with your research activities.

Author Response

We are grateful for all your comments and suggestions.
It´s not clear, why the manuscript contains a detailed description of B. michalskii. A rationale should be provided in the introduction. 
We have two reasons to update the B. michalskii morphological descriptions. First, we found the isolate far from the type and the second locality and associated with quite different host and vector species. It shows that the parasite association is highly flexible and ready to expand in new areas thus creating new social risks to kill wooden plants dominated in local regions (e.g. Dagestan, the Caspian area with surrounding Azerbaijan, Iran, Turkmenistan and Kazakhstan). We stress that this species maintain its host and vector specificity AT THE GUNUS LEVEL (Ulmus-host and Scolytus-vector in Poland and Dagestan). However, in morphology we found several differences of our isolate and the original description, namely in characters of species and even genus level (median bulb and stylet structure). To prove that we found exactly the B. michalskii we have to describe it in a way as we see this species with the detailed morphometrics. 
In addition, the manuscript is lacking a perspective, how species diversity can be captured nowadays. There are NGS approaches published that provide e.g. insights into nematode diversity in deadwood. Hence, it should be discussed why many of your isolated individuals are not determined/ identified to the species level. What are the challenges?

We share your willing to identify all collected isolates to species level. E.g. ‘tylenchid’ sp1 and  ‘tylenchid sp. 2 are evidently Nothotylenchus spp which was confirmed by the cladogram in the Supplement (Tylenchina tree). However if we keep the INTEGRATIVE TAXONOMY way of analysis, we need to study both morphological characters (adults and dauers as the taxonomically functional objects) as well as molecular data; the latter are restricted by the available molecular databases. Here we meet two obstacles that restrict our wishes to identify species. First, we rarely obtain the adult stages of nematodes from saproxylic insects, mostly dauers, and dauers only rarely can be propagated in the lab cultures up to taxonomically informative adult individuals. Second, the molecular databases are far to be complete to identify all nematodes which we collected. Thus we identify our materials to the level of technology tools and data at our disposal with a perspective to use obtained data if the detected nematode isolates will expand and create the problems to society in regions. Of course, it is our challenge, yours and ours.

Abstract ln 20: elytrae of insects.
= elytrae is removed in this place according to proposal of Reviewer-1.

It´s B. fraudulentus not fradulentus (this is misspelled in the entire manuscript)
Corrected

Ln 31: refugia instead of refuges
Corrected 
Ln 45: collected in Russia and Belarus!
Corrected

Lns 51-53: I don´t understand this sentence, rephrase.
Corrected
Ln 160: I would move Blandford to line 156 (first mentioning of the species)
Corrected

Lns 156ff: the entire paragraph contains typos, e.g. it´s one sequence and not sequences…
Corrected

Ln 322: Ulmaceae
 Corrected

Ln 356: ITS
Corrected as ‘Percentage of its identity’

Ln 450: I would write wood-boring Curculionidae including Scolytinae or only write Scolytinae
Corrected!

Ln 452: bark and dauers? -> dauer juveniles
Corrected

Last: Sequences not available yet, I checked at GenBank. Please make sure they are accessible upon publication.
 Sequences will be released after manuscript publication.

Thanks!
Sincerely
Sergei Subbotin and Alexander Ryss